# *Bifidobacterium* Is Enriched in Gut Microbiome of Kashmiri Women with Polycystic Ovary Syndrome

**DOI:** 10.3390/genes13020379

**Published:** 2022-02-18

**Authors:** Saqib Hassan, Marika A. Kaakinen, Harmen Draisma, Liudmila Zudina, Mohd A. Ganie, Aafia Rashid, Zhanna Balkhiyarova, George S. Kiran, Paris Vogazianos, Christos Shammas, Joseph Selvin, Athos Antoniades, Ayse Demirkan, Inga Prokopenko

**Affiliations:** 1Section of Genetics and Genomics, Department of Metabolism, Digestion and Reproduction, Imperial College London, London SW7 2AZ, UK; s.hassan@imperial.ac.uk (S.H.); m.kaakinen@surrey.ac.uk (M.A.K.); h.draisma@surrey.ac.uk (H.D.); z.balkhiyarova@imperial.ac.uk (Z.B.); 2Department of Microbiology, School of Life Sciences, Pondicherry University, Puducherry 605014, India; josephselvinss@gmail.com; 3Section of Statistical Multi-Omics, Department of Clinical and Experimental Medicine, University of Surrey, Guildford GU2 7XH, UK; l.zudina@surrey.ac.uk (L.Z.); a.demirkan@surrey.ac.uk (A.D.); 4Department of Endocrinology, Sheri-Kashmir Institute of Medical Sciences (SKIMS), Srinagar 190011, India; ashraf.endo@gmail.com (M.A.G.); aaffia.rashid@gmail.com (A.R.); 5Department of Food Science and Technology, School of Life Sciences, Pondicherry University, Puducherry 605014, India; seghalkiran@gmail.com; 6Stremble Ventures Ltd., Limassol 4710, Cyprus; paris.vogazianos@stremble.com (P.V.); athos.antoniades@stremble.com (A.A.); 7AVVA Pharmaceuticals Ltd., Limassol 4710, Cyprus; c.shammas@avvapharma.com; 8Department of Genetics, University Medical Center Groningen, 9713 GZ Groningen, The Netherlands; 9Laboratory UMR 8199-EGID, Institut Pasteur de Lille, CNRS, University of Lille, F-59000 Lille, France

**Keywords:** gut microbiome, hormone, amenorrhea, endocrine, PCOS, Indian population

## Abstract

Polycystic ovary syndrome (PCOS) is a very common endocrine condition in women in India. Gut microbiome alterations were shown to be involved in PCOS, yet it is remarkably understudied in Indian women who have a higher incidence of PCOS as compared to other ethnic populations. During the regional PCOS screening program among young women, we recruited 19 drug naive women with PCOS and 20 control women at the Sher-i-Kashmir Institute of Medical Sciences, Kashmir, North India. We profiled the gut microbiome in faecal samples by 16S rRNA sequencing and included 40/58 operational taxonomic units (OTUs) detected in at least 1/3 of the subjects with relative abundance (RA) ≥ 0.1%. We compared the RAs at a family/genus level in PCOS/non-PCOS groups and their correlation with 33 metabolic and hormonal factors, and corrected for multiple testing, while taking the variation in day of menstrual cycle at sample collection, age and BMI into account. Five genera were significantly enriched in PCOS cases: *Sarcina*, *Megasphaera*, and previously reported for PCOS *Bifidobacterium*, *Collinsella* and *Paraprevotella* confirmed by different statistical models. At the family level, the relative abundance of *Bifidobacteriaceae* was enriched, whereas *Peptococcaceae* was decreased among cases. We observed increased relative abundance of *Collinsella* and *Paraprevotella* with higher fasting blood glucose levels, and *Paraprevotella* and *Alkalibacterium* with larger hip, waist circumference, weight, and *Peptococcaceae* with lower prolactin levels. We also detected a novel association between *Eubacterium* and follicle-stimulating hormone levels and between *Bifidobacterium* and alkaline phosphatase, independently of the BMI of the participants. Our report supports that there is a relationship between gut microbiome composition and PCOS with links to specific reproductive health metabolic and hormonal predictors in Indian women.

## 1. Introduction

PCOS is a common endocrine condition affecting women of reproductive age, characterized by hyperandrogenism, oligo- or amenorrhea, and polycystic ovaries on transabdominal ultrasonography. The worldwide prevalence of PCOS among women of fertile age is about 6–18%, and it varies with the use of different diagnostic criteria, such as Rotterdam criteria, 2003, National Institutes of Health (NIH), and the Androgen Excess Society (AES) criteria, 2006 [1,2,3]. In India, PCOS affects the life of an estimated 6.5 to 19.4 million women of reproductive age, and the prevalence is increasing in parallel with the obesity epidemic [4], while the studies addressing the importance of gut dysbiosis/health in this index population are completely lacking. Indian women are reported to have a high prevalence of PCOS [5] and Indian women with PCOS have higher fasting insulin levels and greater insulin resistance compared with white women with PCOS [6]. Moreover, the prevalence of PCOS is even higher among Kashmiri women and is among the highest in a published series globally [7]. In this regard, at many places in India including Kashmir, screening studies are being carried out on adolescent girls and young women for early diagnosis of PCOS and treatment [8,9,10].

Initially defined as a gynecological disorder [11], PCOS is associated with a constellation of metabolic conditions, such as obesity, dyslipidaemia, metabolic syndrome, endothelial dysfunction, inflammation, insulin resistance, hypertension and other cardiovascular risks [12,13,14,15,16,17,18] in addition to infertility, pregnancy complications and depression [19,20,21]. The development of PCOS is multifaceted and involves genetic [22,23], gestational, environment [24] and lifestyle aspects [25]. The precise factors responsible for these key biochemical and metabolic derangements affecting external environment and internal ecosystems, such as gut microbiota, remain largely unexplored. From a genetic point of view, the human gut microbiome is a collection of microbial genomes of microorganisms that inhabit the human gut [26]. The human gut microbiome is a complex ecosystem harbouring numerous microbes taking part in essential functions of the host organism [27]. Earlier case-control studies in Europeans [28] and Chinese [29] showed differences in diversity and relative abundance of particular taxa and their correlation to hormonal levels and body mass index (BMI). The relative abundances of *Bacteroides*, *Escherichia/Shigella* and *Streptococcus* were inversely correlated with ghrelin, and positively correlated with testosterone and BMI in Chinese women with PCOS. Overall, relative abundances of *Akkermansia* and *Ruminococcaceae* are inversely correlated with body weight, sex-hormone, and brain–gut peptides, and are decreased in PCOS [29]. The gut microbiota alterations in humans are associated with obesity [30,31], which may be the driving factor for some of the links with PCOS. Additionally, the peripheral insulin sensitivity in metabolic syndrome subjects improves upon transfer of stool from healthy donors, suggesting a causal effect of gut microbiome on metabolism [32]. Moreover, gut microbiota and its metabolites can control inflammatory processes, brain gut peptide secretion as well as islet b-cell proliferation. Hence, changes in microbiota composition may lead to excessive accumulation of fat ultimately causing insulin resistance and compensatory hyperinsulinemia [32,33]. Gut microbiota also associate with hormonal outcomes. In a female mouse model, the testosterone levels increased upon infecting it with male faecal microbiota compared to unmanipulated females [34]. A host’s estrous cycles, sex hormones and morphological changes in ovaries are affected by gut microbial composition of the host [35]. In rats, the pre-natal exposure to high androgen levels in daughters from mothers with PCOS led to dysbiosis in gut microbiome and impairment of the cardiometabolic functions [36]. The association between gut microbiome, obesity and host genetics suggests that certain bacteria predisposing to a healthy or unhealthy metabolic state may be partially heritable [37,38,39]. It is also hypothesized that the diet might induce bacterial dysbiosis leading to inflammation, insulin resistance and hyperandrogenemia in PCOS [40]. Gut microbiota also play a prominent role in thyroid dysfunction of the host [41], which is a comorbid condition to PCOS [42]. As PCOS is a complex multi-faceted disease involving multiple systems of a woman’s body and it also has a strong environmental component, we questioned to what extent the gut microbiome is involved and whether such involvement can be explained by, e.g., obesity, hormonal levels, and insulin resistance.

We profiled the gut microbiome of young women with PCOS and healthy controls from Kashmir, Northern India, in order to address this understudied population with high PCOS incidence. We used 16S rRNA profiling and further tested whether the gut microbiome composition associates with blood biochemistry and hormonal levels.

## 2. Materials and Methods

### 2.1. Study Cohort and Recruitment

Twenty women with PCOS (drug naive) and 20 control women without PCOS, both groups in age ranging 16–25 years, were identified during regional PCOS screening of college and school students in different schools and colleges in Kashmir, North India, from January to May 2017. Those women with regular cyclicity in their menstrual cycles (21–35 days), no signs of hyperandrogenism and that had normal ovarian morphology as evidenced through transabdominal ultrasonography were invited to take part as controls. The case group consisted of participants with menstrual disturbances including oligomenorrhea (menstrual interval > 35 days or <8 cycles/year) or amenorrhea (no menstrual cycle in last >6 months), hyperandrogenism (male pattern hair growth, androgenic alopecia), polycystic ovaries on transabdominal ultrasonography and qualified Rotterdam criteria, 2003 [1] for PCOS diagnosis. According to the Rotterdam criteria 2003, a clinical diagnosis of PCOS requires that a patient presents with two of the following symptoms: (i) oligo-ovulation or anovulation, (ii) Hyperandrogenism- (including signs such as hirsutism), and (iii) polycystic ovaries visible on ultrasound. Hirsutism was assessed by Ferriman–Gallwey [43]. In total, nine body parts, including the upper lip, chin, chest, upper and lower abdomen, thighs, upper and lower back, and upper arms, were examined for the presence of terminal hair. Terminal hairs at these sites were graded in the range of 0–4. The Ferriman–Gallwey score was determined as the cumulative score of all the sites. Hirsutism was defined by the presence of a modified Ferriman–Gallwey score of 8 or higher.

Initially, among the individuals who gave informed consent during the interview, 32 were selected as non-PCOS and 53 selected as PCOS and were invited to donate blood and stool samples on days 3–7 (early follicular phase) of spontaneous cycle to maximise hormonal homogeneity. Tests for follicle stimulating hormone (FSH), luteinising hormone (LH) and testosterone by standard are best performed during the follicular phase (3 to 7 days of the menstrual cycle.) The blood and stool donation took place within the same day at the Endocrine clinic of the Sher-i-Kashmir Institute of Medical Sciences (SKIMS, a tertiary care hospital). Eventually, 23 individuals without and 20 individuals with PCOS donated both blood and stool samples. Three samples from the control group were excluded later as they did not fit in the required follicular phase.

Women on antibiotic treatment or those taking contraceptives, steroids, anti-epileptics, insulin sensitizers, proton-pump inhibitors, non-steroid anti-inflammatory drugs, or with any previous history of systemic sickness such as diabetes mellitus, coronary artery disease, non-classical congenital adrenal hyperplasia (NCAH), Cushing syndrome, hyperprolactinemia, thyroid dysfunction, gastrointestinal disease and appendectomy were excluded from the study. All participants were non-vegetarian. One person from the PCOS group was later excluded due to low sequencing read depth. The study protocol was approved by the Institutional Ethics Committee (IEC) of SKIMS Kashmir and written informed consent was obtained from all the participants involved in this study.

### 2.2. Blood Biochemistry, Hormonal Tests and Anthropometry

Fasting blood samples were collected in dry and EDTA coated tubes after 10–12 h fasting. Sera were separated to be used in measuring glucose, lipids, alkaline phosphatase, aspartate aminotransferase, albumin and creatinine, in addition to hormones (prolactin, thyroid stimulating hormone (TSH), thyroxine (T4), triiodothyronine (T3), cortisol, luteinising hormone (LH), follicle stimulating hormone (FSH), and total testosterone. The EDTA-containing aliquot was immediately placed on ice and centrifuged within 30 min; plasma was collected and was stored at −80 °C for further analysis. Alkaline phosphatase (ALP), aspartate aminotransferase (AST), creatinine, albumin and lipid measurements were performed using a fully automated chemistry analyser (Hitachi 912 at SKIMS). Estimation of serum T3, T4, cortisol, TSH, prolactin, LH, FSH and total testosterone was done by Radioimmunoassay (RIA) using commercial kits and according to the supplier protocol at SKIMS. Plasma glucose was measured by the glucose oxidase peroxidase method. Body weight (light clothing without shoes), height and waist circumference were measured with standard calibrated instruments (SECA 213, Hamburg, Germany), followed by a detailed systemic examination that included blood pressure measurement (Omron HEM7120).

### 2.3. Stool Samples

Each participant was asked to provide fresh stool sample (approx. 5 g) at the same day with blood collection, by using the stool collection and stabilization kit (OMNIgene^®^•GUT OMR-200, DNA Genotek, Canada). All samples were collected in the morning and were stored at ambient temperature until further processing which took place within the next 60 days. Consistency of each sample was recorded on the following scale: 1 = Sausage shaped with cracks on the surface, 2 = Sausage shaped and smooth soft stool, 3 = Solid stool with clumps, 4 = Watery stool.

### 2.4. Metagenomic DNA Extraction and 16S rRNA Sequencing Data Generation

DNA extraction was performed by as per the instruction manual using ZymoBIOMICS^TM^ DNA kit by Zymo Research USA. The DNA concentrations were estimated by Qubit Fluorometer (Thermo Fisher) and checked by Agilent TapeStation 2200. The microbiota characterization was performed by targeting the hypervariable regions V3–V4 of 16S rRNA gene using a paired-end approach using the specific primers published earlier [44] and according to the manufacturer’s instructions [45]. The amplified regions were combined with dual-index barcodes, Nextera^®^ XT Index Kit v2 Set A, B and C, Illumina (Illumina Inc., San Diego, CA, USA) The sequencing run was performed with MiSeq 600 cycle Reagent Kit v3, Illumina USA and sequenced on a MiSeq Illumina system by the Stremble Ventures LTD, Cyprus.

### 2.5. Bioinformatic Analysis and Quality Control

The bioinformatic analysis including read QC and operational taxonomic unit (OTU) classification was performed from FASTQ files with paired-end reads. Specifically, the Basespace platform and the 16S Metagenomics from Basespace (Illumina recommended software) were used to deal with quality filtering and denoising of the raw reads. This also classified OTUs using Ribosomal Database Project Classifier [46] against the Illumina-curated version of the GreenGenes reference taxonomy database [46,47]. The number of reads classified at the genus and family level are given in Table 1. Resulting Krona plots of all taxonomic entities detected are provided in Appendix A. Among those, genera and families that were detected in more than 1/3 of the samples were considered prevalent OTUs and, among those, the ones with >0.1% average relative abundance across all subjects were included in analysis (Appendix A).

### 2.6. Statistical Analysis

#### 2.6.1. Association Analyses between Individual Species, PCOS and Hormone Levels

We performed all the analyses at Family and Genus levels for which the proportion of successfully classified reads across samples was over 95%. We used the non-parametric Mann–Whitney U-test [48] to compare median abundancies of each species between PCOS cases and controls. For these analyses, we considered results with a *p*-value < 0.05 as statistically significant. To account for potential confounding factors, we used the Multivariate Association with Linear Model, MaAsLin, in R [49], which allows for the use of covariates in the model. In brief, MaAsLin fits a linear model for each species and the variable of interest, here PCOS, after arcsin-square-root transformation of the proportional values of the species. This transformation has been shown to stabilise variance and normalise proportional data well [49]. The method can also include a boosting step to select factors among a large set of variables to be associated with the species. We turned this feature off since we focused only on one variable of interest, PCOS. We used the default settings of the programme, i.e., minimum relative abundance of 0.01%; minimum percentage of samples in which a feature must have the minimum relative abundance in order not to be removed of 10%; outlier removal by a Grubbs test with the significance cut-off used to indicate an outlier at 0.05; multiple test correction by the Benjamini–Hochberg (BH) [50] method; and the threshold to use for significance for the generated q-values (BH FDR) of 0.25. As potential confounders in the model, we used stool consistency, read depth, age, day of the menstrual cycle during sample collection and BMI. The linear model analysis with MaAsLin was performed also for LH, FSH, and testosterone levels in cases and controls together, adjusting for stool consistency, read depth, age, day of the menstrual cycle during sample collection and BMI. For bacteria abundances that were different between cases and controls, we performed follow-up analyses with a range of quantitative traits measured from the participants, such as blood glucose and lipid levels (Appendix A), to dissect the underlying mechanistic links between the identified OTUs and PCOS. We used linear regression by MaAsLin and the same adjustments as before. The analyses with hormonal variables and other quantitative traits for those family and genera that showed a statistically significant association were also performed separately in PCOS cases and controls.

#### 2.6.2. Microbial Distance and α Diversity

Inter-individual microbial distance (Bray–Curtis dissimilarity) and α diversity (Shannon’s diversity index) were calculated using functions *vegdist* and *diversity* of the R package *vegan* [51]. We analysed the association between PCOS and (i) Bray–Curtis dissimilarity and (ii) Shannon’s diversity index using Spearman correlation in R. Both analyses were additionally adjusted for stool consistency, day of menstrual cycle at sample collection and sequencing read depth.

## 3. Results

### 3.1. 16 SrRNA Gut Microbiome in PCOS

In this study, we initially profiled the microbiome in DNA isolated from faecal samples by 16S rRNA sequencing in 20/20 women with/without PCOS from Kashmir, India. After the quality control of 16S rRNA sequencing data, and removing one person from the case group due to low number of reads, we analysed the 39 microbiomes of Kashmiri women with/without PCOS (Table 1 and Appendix A). The Krona plots (Appendix A) and the bar plots, (Figure 1 and Appendix A) show that the core phyla of gut microbiomes are mainly dominated by *Firmicutes* and *Bacteroidetes* followed by *Proteobacteria* in controls (Appendix A) and *Actinobacteria* in patients (Appendix A). We defined 40 OTUs at the family level and 58 at the genus level (Appendix A). These were detected in at least 30% of the samples with relative abundance > 0.1%.

### 3.2. Associations with PCOS at the Genus and Family Level

The analyses of the taxa at genus level showed statistically significant (*p* < 0.05) differences when compared between individuals with and without PCOS for 12 genera either by the Mann–Whitney U-test or by linear modelling with MaAsLin (Table 2). Ten of these bacterial genera reached false discovery rate (FDR) corrected statistical significance (Q < 0.25) in the linear modelling with arcsin-square root transformation and outlier exclusions (Table 2). The most striking differences of relative abundance were observed for *Sarcina* (cases vs. controls: 0.28% vs. 0.06%), *Megasphaera* (3.62% vs. 1.17%) and *Bifidobacterium* (7.7% vs. 3.1%) (Figure 2). After adjustment by stool consistency, day of menstrual cycle at data collection point, sequencing read depth, age and BMI (Model2) the relative abundance of *Sarcina, Megasphaera, Collinsella, Paraprevotella*, and *Bifidobacterium* associated with PCOS, consistently in all three statistical models performed (Table 2 and Appendix A, Figure 2). It is worth noting that, although *Alkalibacterium* associated with case/control status in all of the three models, the direction of association by the Mann–Whitney U-test was the opposite of what was estimated by MaAslin. Therefore, we considered it as a false positive.

At the family level, increased relative abundance of *Bifidobacteriaceae* associated with case status (7.63% vs. 3.07%), whereas relative abundance of *Aerococcaceae* (0.08% vs. 0.31%) and *Peptococcaceae* (0.15% vs. 0.27%) were decreased among individuals with PCOS (Figure 2, Appendix A) when analysed by our Model 2, taking the BMI differences into account. However, only *Peptococcaceae* and *Bifidobacteriaceae* showed consistency between the Mann–Whitney U-test and all three MaAsLin models and were considered further. Figure 3 shows the Shannon diversity index between case and control groups.

### 3.3. Hormonal Profiles at the Genus and Family Level

Upon linear modelling of the FSH, LH, LH to FSH and testosterone hormonal profiles for all study participants, *Eubacterium* reached the FDR-corrected significance level for association with FSH (coeff = 0.15, *p* = 0.0003, Q = 0.023, Appendix A, Figure 4a). This association remained significant after adjustment for the covariates (Appendix A). We further checked the association separately in cases and controls and observed statistically significant associations in both groups, in the same direction (Cases: coeff = 0.016, *p* = 0.022; Controls: coeff = 0.017, *p* = 0.017). For the other hormones, we did not observe statistically significant associations for any bacterial species (Appendix A). Linear regression analysis of hormonal profiles at family level relative abundancies did not yield any statistically significant associations (Appendix A).

### 3.4. Follow-Up of the Identified OTUs at the Genus and Family Level

The study participants were assessed for a number of quantitative traits (Table 1 and Appendix A) enabling us to test the supporting hypothesis of involvement of gut microbiome with continuous biochemical and hormonal markers of PCOS constellation. At genus level (Table 2), the quantitative trait analyses showed an association between higher *Bifidobacterium* and higher alkaline-phosphatase (ALP) level, which remained significantly associated after further adjustment by BMI (coeff = 0.0011, *p* = 0.00019, Q = 0.214), (Appendix A, Figure 3b). We were statistically underpowered to detect any associations separately in cases and controls. Among the remaining genera from Table 2, increased abundance of *Lactobacillus* (coeff = 0.001, *p* = 0.0004, Q = 0.156), driven by PCOS cases (Cases: coeff = 0.001, *p* = 0.004; Controls: coeff = 0.00004, *p* = 0.91) and *Collinsella* (coeff = 0.0004, *p* = 0.002, Q = 0.199), also driven by PCOS cases (Cases: coeff = 0.0004, *p* = 0.03; Controls: coeff = 0.0002, *p* = 0.32) associated with higher ALP, whereas *Sarcina* (coeff = −0.001, *p* = 0.002, Q = 0.204) negatively associated with another liver enzyme measured, aspartate transaminase (AST). Additionally, *Paraprevotella* associated with fasting glucose (coeff = 0.00077, *p* = 0.0014, Q = 0.170), waist (coeff = 0.00073, *p* = 0.0010, Q = 0.170) and hip circumference (coeff = 0.00065, *p* = 0.0028, Q = 0.204), as well as weight (coeff = 0.0007, *p* = 0.002, Q = 0.199). Increased abundance of *Alkalibacterium* associated with waist (coeff = 0.00034, *p* = 0.0014, Q = 0.170) and hip circumference (coeff = 0.00031, *p* = 0.0025, Q = 0.204) and weight (coeff = 0.00037, *p* = 0.0017, Q = 0.186) (Appendix A). At the family level, *Bifidobacteriaceae* (coeff = 0.0011, *p* = 0.00012, Q = 0.192) and *Lactobacillaceae* (coeff = 0.0010, *p* = 0.00050, Q = 0.192) associated with increased ALP, the latter driven by the association in PCOS cases (Cases: coeff = 0.0011, *p* = 0.004; Controls: coeff = 0.00005, *p* = 0.91), and *Peptococcaceae* with decreased prolactin (coeff = −0.0011, *p* = 0.00042, Q = 0.192) (Appendix A). None of these associations were significant after BMI adjustment (Appendix A), except the one between *Bifidobacterium* and ALP.

## 4. Discussion

This is the first investigation of PCOS gut microbiome in women from Kashmir, India where incidence of PCOS is high and comorbid with greater insulin resistance. Here, we compared 39 individuals with/without PCOS by their gut microbiome composition and dissected the latter in relation to 33 quantitative endophenotypes. We performed a step-wise association modelling moving from simplistic towards more complex models with covariates. We identified robust signals both at genus and family level taxonomy. Seven taxa were significantly different in PCOS cases, including enrichment at both genus and family levels, i.e., relative abundance of *Bifidobacterium* and *Bifidobacteriaceae*. When considering all samples together, we observed the domination by *Firmicutes* followed by *Bacteroidetes**, Actinobacteria* and *Proteobacteria*, the same enterotype reported by Das et al. [52] for samples collected from two distinct regions of India, including Leh, located in proximity of Kashmir. Overall, the gut microbiome of women with PCOS had a higher bacterial diversity compared to that of women without PCOS, when measured by Shannon’s index. This is a unique finding as previous studies showed decreased diversity among individuals with PCOS as compared to without [53,54] and our observation may be specific to the Kashmiri population. At the genus level, the relative abundances of five genera including *Lactobacillus, Bifidobacterium, Sarcina, Megasphaera, Collinsella* and *Paraprevotella,* were increased in gut microbiome of women with PCOS. Association of *Lactobacillus* seems to be dependent on BMI as tested by our model 2, whereas the other genera remained significantly associated after adjustment for BMI. Among them, *Bifidobacterium* also associated with higher ALP, independent of BMI. We also show that *Collinsella* and *Paraprevotella* are associated with higher fasting blood glucose levels. Our study is the first to report *Sarcina* and *Megasphaera* in relation to PCOS. At the hormonal level analysis of all taxa, we show that increased *Eubacterium* is associated with increased FSH.

At the family level, the gut microbiomes of women with PCOS are enriched with *Bifidobacteriaceae*, whereas they harbour lower relative abundance of the *Peptococcaceae* family when the technical covariates are accounted for in statistical models. The cases had higher mean BMI; however, there was no difference in WHR between two study groups, thus not suggesting higher adiposity as the driver for anthropometric differences (Table 1). We additionally adjusted our tests for variation in BMI, so the associations we report are not likely to be confounded by the BMI of the participants, unless reported otherwise. The individuals with PCOS also had higher blood glucose, prolactin and TSH levels (Table 1 and Appendix A) consistent with earlier reports [55]. It is of note that both fasting glucose and prolactin were increased among our case group and at the same time we saw linear associations between decreased abundance of family *Peptococcaceae* and lower prolactin, and increased abundance of genera *Collinsela* and *Paraprevotella* and fasting glucose, when looking at the whole population. However, these were likely to be driven by BMI. Previous studies have indicated that BMI has an important role in the diversity of the gut microbiome, and that having a higher BMI is strongly linked to gut dysbiosis [56,57,58].

To date, a few observational studies on the gut microbiome of individuals with PCOS exist [26,29,35,54,59]. It has been reported that the probiotic *Bifidobacterium lactis V9* may increase sex hormone levels in PCOS patients [59]; however, we cannot test this at the cross-sectional design as we had limited the taxonomic resolution to genus and family levels. Relative *Bifidobacterium* abundance in human gut is known to be driven by lactose intolerance. In lactose-intolerant individuals, lactose is not metabolized in the small intestine and proceeds to the colon where it is fermented by members of the gut microbiome. This fermentation leads to gas production, a major symptom associated with lactose intolerance [60]. Thus, genetic variants that reduce lactase activity can promote the growth of lactose-fermenting bacteria in the colon, but only if the individual consumes dairy products. Taken together with our findings, there could be a relationship between the adult type hypolactasia and PCOS. There has been only one small study looking at such an association; however, it has not been replicated [61]. The *Bifidobacterium* abundance in gut is known to associate with favorable metabolic outcomes, but a recent report showed that not all strains of *Bifidobacterium* are functional [62]. These strain level differences can only be resolved by metagenomic sequencing, which requires further investigation. We earlier reported that rs182549 (human *LCT* gene)*C allele, predisposing to lactose intolerance, and thus to increased *Bifidobacterium* in gut, was negatively associated with height and obesity and positively associated with several nutritional phenotypes, type 2 diabetes (T2DM) risk and family history of T2DM in the UK Biobank cohort [39]. Moreover, the functional *LCT* SNP rs4988235 variant was in strong linkage disequilibrium associated with 1,5-anhydroglucitol (*p* = 4.23 × 10^−28^), which is an indicator of glycemic variability [39]. In our study, *Bifidobacterium* also associated with ALP in the overall sample; however, the ALP values from individuals with PCOS were higher than of control group, 250.7 vs. 210 mg/dL (*p* = 0.016, by *t*-test). ALP is a main indicator of liver function; however, it is also secreted in the intestine where it plays a vital role in maintaining gut homeostasis [63]. Thus, the link between ALP and *Bifidobacterium* could be mediated through liver function, or could be attributed to direct influence of colonic ALP on the gut homeostasis. It is of note that increased prevalence of non-alcoholic fatty liver disease (NAFLD) and hepatic steatosis were reported in patients with PCOS [64,65,66]. It is hypothesised that alcohol producing bacteria such as *Bifidobacterium* may contribute to the pathogenesis of NAFLD in PCOS [67,68]. Another possible link could be via primary bile acids like glyco- and tauro-conjugated which were found to be elevated in women with PCOS compared to controls and were positively associated with HA [69]. Bile acid metabolism begins in the gastrointestinal tract involving microbiota possessing bile salt hydrolase activity [70], which is common in *Bifidobacterium* and *Lactobacillus* [71]. *Bifidobacterium* is an important genus for host health, associated with favorable outcomes, especially in the early years of life. Taken together with shared genetics and association to glycemic traits, as well as liver functioning, the association of *Bifidobacterium* may have multiple (age-dependent) facades, including unfavorable effects.

*Collinsella* and *Paraprevotella*, which we observed to be associated with fasting glucose, were earlier shown to be enriched in the gut of obese women without PCOS, as compared to non-obese PCOS cases, indicating that these genera could be more specific to obesity than the PCOS cluster of phenotypes [29]. On the other hand, the fact that many PCOS patients also have hyperandrogenism may explain the relationship between PCOS and *Paraprevotella*. The genus has been detected in prenatal androgenized rat models, where *Paraprevotella* was significantly enriched in androgenized rodents [72], thus suggesting a mechanistic link between androgens and *Paraprevotella.*

However, as for *Collinsella*, we see a shared genetic link with genus in larger population based studies; in our recent report on genetics of random glucose in humans, we suggest that *ABO/FUT2* genetic locus potentially orchestrates the correlation between *Collinsella* and the host glucose level [73]. Another piece of supporting evidence from a shared genetic background concerns family *Peptococcaceae;* rs7574352 associated with the family *Peptococcaceae* [39] is located in the intergenic region in the proximity (220kb apart) of *IRF1*, which is involved in insulin resistance and susceptibility to T2DM, also suggested as a risk loci for PCOS [74,75].

Contrary to our findings highlighting *Lactobacillus* enrichment in PCOS cases as driven by BMI, *Lactobacillus* was earlier found enriched in non-obese controls when compared to obese PCOS individuals [29], suggesting the involvement of BMI as a confounder or mediator of the association. A randomized controlled study focusing on metabolic benefits of synbiotics in PCOS reported that *Lactobacillus* intake resulted in lower insulin concentration and insulin resistance and higher insulin sensitivity [59], suggesting a causal role for *Lactobacillus* for PCOS accompanied with high BMI. The genera *Megasphaera,* enriched in PCOS gut microbiome samples, are also involved in lactic acid fermentation but have no established role in the PCOS human gut. *Sarcina* is a member of the family *Clostridiaceae;* the genus *Sarcina ventriculi* is increasingly common Gram-positive coccus recognized in gastric biopsies from patients with delayed gastric emptying [76], one of the clinical symptoms in PCOS.

At the hormonal level analysis of all taxa, we show that increased *Eubacterium* is associated with increased FSH; however, the association was similar both in individuals with and without PCOS and does not provide any insight for the main objective of our study and will not be discussed further.

One recent study on a Westernised (Finnish) population conducted by Lüll et al. [77] has observed statistically significant associations between diversity measures and PCOS-related hormonal and metabolic parameters, such as BMI, sex-hormone binding globulin (SHBG) levels and insulin resistance. In the same study, bacterial diversity indices did not differ significantly between PCOS and controls unless they were stratified by prediabetes status; then, women with PCOS showed a lower Shannon diversity index. On the contrary to what has been shown by Lüll et al., we did not observe differences in *Clostridiales, Ruminococcaceae UCG-002*, and *Clostridiales Family XIII AD3011* group and genus *Dorea*. We cannot exclude the fact that differences could well be due to the dietary and regional lifestyle differences, i.e., Finland vs. Kashmir.

The main limitation of our study is the relatively small sample size, however, comparable to the size of other pilot studies in the field. In order to deal with limited sample size, we used the non-parametric Mann–Whitney U-test as an exploratory analysis. Although this method does not allow to account for covariates, the associations with families *Peptococcaceae, Bifidobacteriaceae* and associations with genera *Sarcina, Megasphaera, Bifidobacterium, Paraprevotella* and *Collinsella* turned out to be robust and confirmed by the MaAsLin approach. In addition, our study might be affected by the lack of resolution, which can be compensated by using shotgun metagenomics sequencing in further research. Our study is focused on gut microbiome composition in PCOS, but we had no information on the saliva microbiome phylogenetic profile, which has lower diversity in PCOS and is altered in relation to clinical symptoms of the disease [78].

The microbiome research in PCOS is a fast-expanding field; however, many pilot studies published for PCOS gut microbiome composition require replication of reported association with specific taxa. The emerging field of faecal microbiota transplantation might become promising in PCOS management, given its first encouraging results in rats, which led to decreased androgen biosynthesis and normalisation of ovarian morphology [35].

In this study, we report an increased abundance of genera involved in lactic acid fermentation in the stool samples of individuals with PCOS. These bacterial genera highlight specific effects of gut microbiota beneficial for human metabolism while the links also point out a shared genetic background between hist metabolism and gut microbiota. However, the data available to us do not allow conclusive inferences about these microorganisms being a cause or consequence of metabolic disturbances in PCOS. Further studies are needed for evaluation, whether abundance in these bacterial genera is specific to Indian women. Our findings show importance of gut microbiota in maintenance of hormonal levels in PCOS and warrant further well-powered research.

## Figures and Tables

**Figure 1 genes-13-00379-f001:**
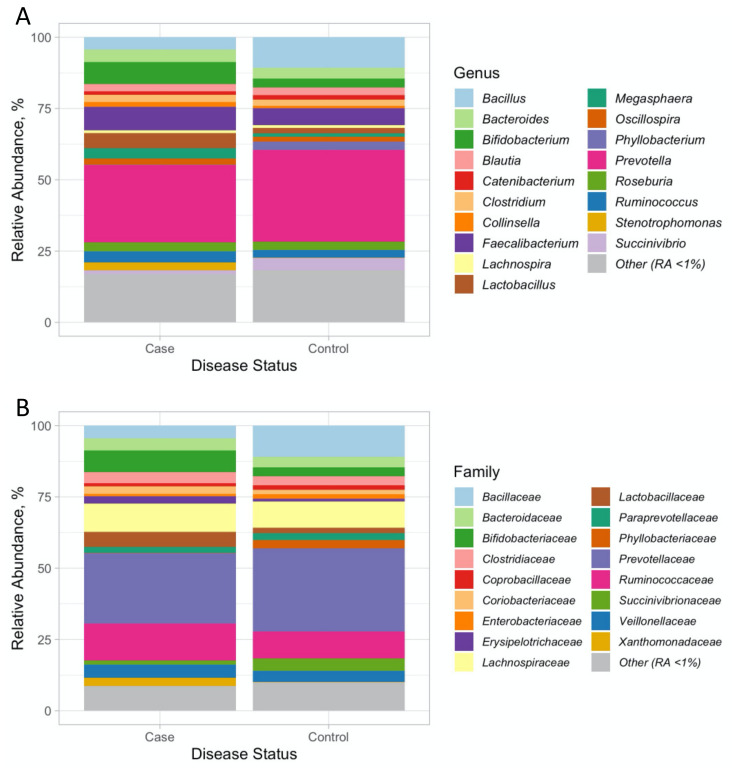
The relative abundancies (RA) of the OTUs at the (**A**) genus and (**B**) family level. For better visualisation, only OTUs with RA > 1% are plotted and the less prevalent ones are grouped together in Other (RA < 1%).

**Figure 2 genes-13-00379-f002:**
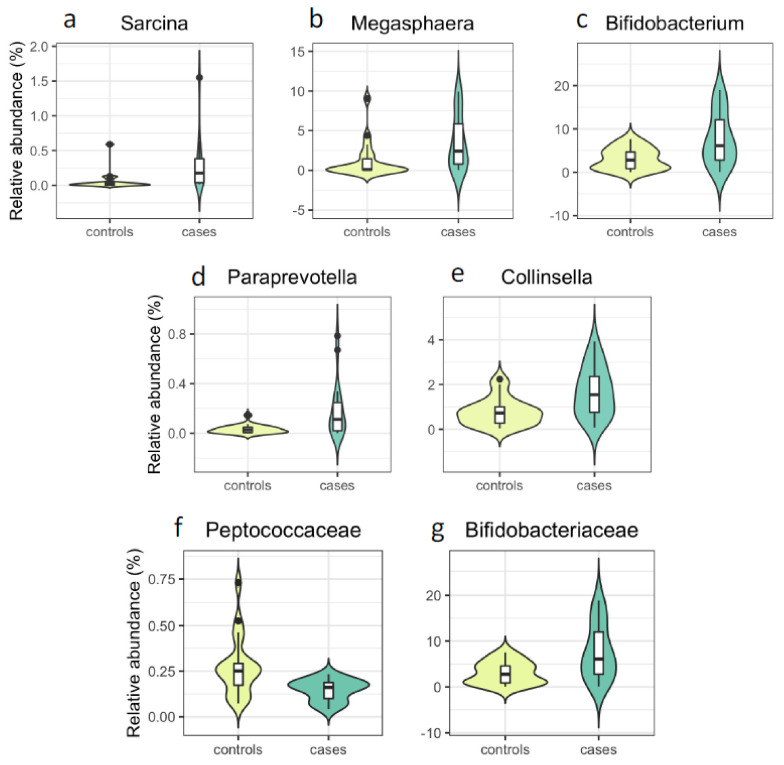
Violin plots showing the distributions of the bacterial genera (**a**–**e**) and families (**f**,**g**), reaching statistical significance consistently in all statistical models from Table 2 and Table 3.

**Figure 3 genes-13-00379-f003:**
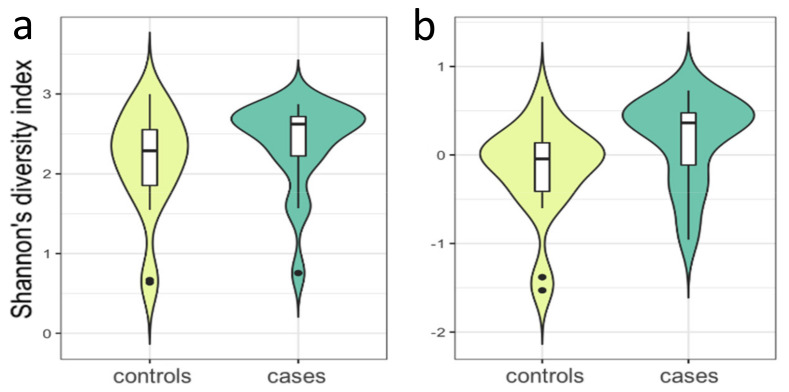
Violin plots showing the distribution of the Shannon diversity index, (**a**) Unadjusted, b: Adjusted for stool consistency, day of menstrual cycle at sample collection and sequencing read depth. The negative values in (**b**) are due to the fact that residuals after adjustment were used in the plots, for a more precise visualisation.

**Figure 4 genes-13-00379-f004:**
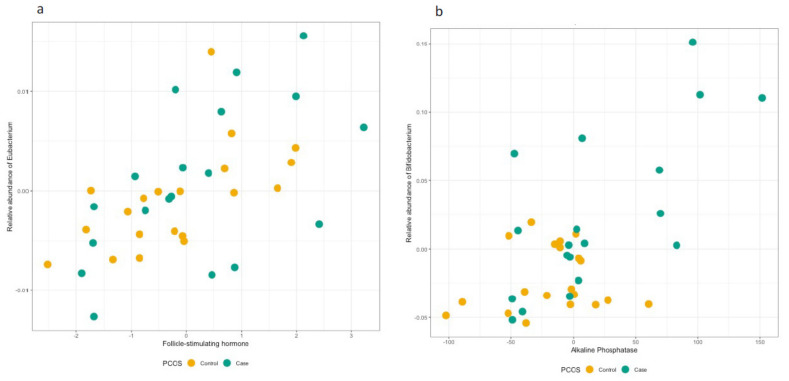
Scatter plots of the relative abundances of (**a**) *Eubacterium* according to FSH levels, and (**b**) *Bifidobacterium* according to ALP levels. Individuals with PCOS are depicted with green, and controls with orange.

**Table 1 genes-13-00379-t001:** Characteristics of the study sample.

	Cases (*n* = 19)	Controls (*n* = 20)	*p*-Value *
	Mean (SD) or *n* [%]	Mean (SD) or *n* [%]	
Anthropometrics and blood measurements			
Age, years	23.9 (6.9)	21.1 (2.5)	0.093
BMI, kg/m^2^	**25.4 (3.3)**	**23.2 (3.2)**	**0.041**
Waist–hip ratio	0.9 (0.1)	0.9 (0)	0.73
Systolic blood pressure, mmHg	124.5 (9.7)	120.8 (6.1)	0.16
Diastolic blood pressure, mmHg	81.6 (7.6)	80.5 (3.9)	0.58
Hormonal and Biochemical Parameters			
Luteinizing hormone (LH), IU/L	5 (2.8)	4.3 (0.6)	0.28
Follicle stimulating hormone (FSH), IU/L	5.5 (1.6)	5.1 (1.2)	0.34
LH/FSH ratio	0.97 (0.70)	0.90 (0.28)	0.66
Testosterone, ng/dL	56.8 (26.6)	49.6 (12.7)	0.28
Prolactin, ng/dL	**15.7 (7.2)**	**10 (4.7)**	**5.64 × 10^−3^**
Fasting blood glucose, mg/dL	**114.2 (8.6)**	**106.1 (9.1)**	**6.71 × 10^−3^**
Blood glucose 1 h, mg/dL	126 (11.8)	120 (24.3)	0.33
Blood glucose 2 h, mg/dL	121.2 (10.7)	121.3 (22.3)	0.99
Total cholesterol, mg/dL	164.7 (31.8)	149.1 (26.9)	0.11
Triglycerides, mg/dL	205.9 (105.8)	144.3 (96.6)	0.065
High-density lipoprotein (HDL), mg/dL	39.6 (4.6)	41.8 (4.4)	0.14
Low-density lipoprotein (LDL), mg/dL	110.2 (8.4)	108.4 (9.5)	0.54
PCOS-related clinical criteria			
Age at menarche, years	13.3 (2.2)	12.2 (1.2)	0.063
Number of cycles per year	6.8 (1.9)	10.7 (1)	1.69 × 10^−9^
Hirsutism score	7.3 (5.5)	0.4 (1.8)	6.03 × 10^−6^
Menstrual cycle irregularity, yes	19 [100]	0 [0]	1.45 × 10^−11^
Acanthosis nigricans, yes	5 [26]	0 [0]	0.02
Acne, yes	7 [37]	2 [10]	0.06
Alopecia, yes	9 [47]	4 [20]	0.10
Duration of hirsutism, years	1.8 (0.8)	0.1 (0.2)	1.87 × 10^−11^
Family history of hirsutism, yes	3 [16]	2 [10]	0.66
Family history of menstrual disturbances, yes	5 [26]	2 [10]	0.24
Family history of T2D, yes	11 [58]	7 [35]	0.2
Stool sample collection and sequencing			
Day of menstrual cycle at stool collection			
Day 3	7 [37]	6 [30]	1.00
Day 4	5 [26]	6 [30]	
Day 5	6 [32]	7 [35]	
Day 6	1 [5]	1 [5]	
Stool consistency			
Sausage shaped with cracks on the surface	3 [16]	2 [10]	0.66
Sausage shaped and smooth soft stool	12 [63]	15 [75]	
Solid clumpy stool	3 [16]	1 [5]	
Watery stool	1 [5]	2 [10]	
Read depth, genus level	80,434.6 (44,927.2)	161,279.9 (158,861.1)	0.039
Read depth, family level	81,547.8 (45,539.7)	164,695 (162,499.2)	0.038

* *p*-Value is from *t*-test for continuous traits and Fisher’s exact test for categorical traits.

**Table 2 genes-13-00379-t002:** Statistically significant differences between PCOS cases and controls at the genus level.

	Mann–Whitney U-Test			MaAsLin									
				Unadjusted				Model 1			Model 2		
Feature	Case	Control	*p*-Value	*n*	Coefficient	*p*-Value	Q-Value	Coefficient	*p*-Value	Q-Value	Coefficient	*p*-Value	Q-Value
*Sarcina*	0.28	0.059	3.4 × 10^−4^	36	0.023	0.001	0.025	0.023	0.001	0.066	0.024	0.001	0.127
*Megasphaera*	3.62	1.17	2.1 × 10^−3^	39	0.093	0.003	0.048	0.099	0.003	0.104	0.098	0.005	0.190
*Bifidobacterium*	7.72	3.12	7.5 × 10^−3^	39	0.096	0.006	0.048	0.100	0.004	0.115	0.103	0.005	0.190
*Paraprevotella*	0.18	0.032	7.3 × 10^−3^	37	0.014	0.004	0.048	0.014	0.001	0.066	0.012	0.006	0.192
*Collinsella*	1.68	0.78	7.5 × 10^−3^	39	0.041	0.004	0.048	0.043	0.002	0.101	0.043	0.004	0.179
*Erysipelothrix*	1.52	0.29	7.5 × 10^−3^	39	0.017	0.036	0.219	0.018	0.022	0.342	0.016	0.058	0.475
*Lactobacillus*	5.30	1.90	0.012	39	0.086	0.006	0.048	0.085	0.008	0.207	0.084	0.014	0.304
*Dysgonomonas*	0.66	0.18	0.013	39	0.016	0.035	0.219	0.016	0.047	0.481	0.016	0.063	0.501
*Oscillospira*	2.22	1.68	0.018	39	0.033	0.168	0.454	0.039	0.06	0.487	0.036	0.104	0.574
*Natronincola*	0.35	0.31	0.030	38	0.016	0.049	0.256	0.017	0.016	0.298	0.016	0.031	0.456
*Alkalibacterium*	0.089	0.32	0.049	35	0.007	0.001	0.025	0.007	0.000	0.066	0.006	0.002	0.145
*Atopobium*	0.27	0.08	0.066	36	0.020	0.038	0.219	0.019	0.042	0.481	0.019	0.057	0.475

Mean: (%) in cases (*n* = 19) and (%) in controls (*n* = 20). Model 1: Adjusted for stool consistency, read depth, day of menstrual cycle at sample collection, age. Model 2: Adjusted for stool consistency, read depth, day of menstrual cycle at sample collection, age, BMI.

**Table 3 genes-13-00379-t003:** Statistically significant differences between PCOS cases and controls in the relative abundancies at the family level.

Feature	Mann–Whitney U-Test			MaAsLin									
	Mean (%) in Cases (*n* = 19)	Mean (%) in Controls (*n* = 20)	*p*-Value	Unadjusted				Model 1			Model 2		
				*n*	Coefficient	*p*-Value	Q-Value	Coefficient	*p*-Value	Q-Value	Coefficient	*p*-Value	Q-Value
*Peptococcaceae*	0.15	0.27	1.5 × 10^−3^	39	−0.011	0.003	0.055	−0.010	0.005	0.246	−0.010	0.005	0.233
*Bifidobacteriaceae*	7.63	3.07	7.5 × 10^−3^	39	0.096	0.005	0.055	0.100	0.003	0.233	0.103	0.005	0.233
*Lactobacillaceae*	5.26	1.89	0.012	39	0.086	0.005	0.055	0.085	0.008	0.307	0.083	0.014	0.349
*Erysipelotrichaceae*	2.50	0.91	0.026	39	0.030	0.034	0.254	0.029	0.045	0.457	0.022	0.124	0.680
*Porphyromonadaceae*	0.88	0.43	0.047	39	0.015	0.058	0.254	0.016	0.051	0.461	0.015	0.094	0.680
*Aerococcaceae*	0.08	0.31	0.052	35	0.007	0.001	0.037	0.007	0.000	0.082	0.006	0.002	0.233

Mean: (%) in cases (*n* = 19) and (%) in controls (*n* = 20). Model 1: Adjusted for stool consistency, read depth, day of menstrual cycle at sample collection, age. Model 2: Adjusted for stool consistency, read depth, day of menstrual cycle at sample collection, age, BMI.

## Data Availability

The datasets generated during and/or analysed during the current study are not publicly available for reasons related to privacy and participant consent but are available from the corresponding author on reasonable request. Summary-statistics level results are included in the Appendix A attached to the manuscript.

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
