# Peer review of "Bifidobacterium* Is Enriched in Gut Microbiome of Kashmiri Women with Polycystic Ovary Syndrome"

_genes, 2022, doi:10.3390/genes13020379_

Round 1
Reviewer 1 Report
In this manuscript, Hassan et al profiled the gut microbiota composition of 19 women with polycystic ovary syndrome (PCOS) and compared those with 20 women with no PCOS. Authors identified several differentially abundant microbial genera and families, and correlated abundances of these taxa with levels of several blood markers and hormones.
The major shortcoming of this manuscript is that it lacks discussion of findings in the context of PCOS. While authors report several associations between gut species, blood markers and hormone levels (sections 3.3 and 3.4), there is inadequate discussion as to how these associations are relevant to PCOS, what is known about the implicated species in the context of PCOS, speculation of probable mechanisms or referencing literature as evidence to support inferences. As such, I am not sufficiently convinced that compositional differences in the gut microbiota of women with PCOS reported here are related to PCOS. It would also be great to see authors further compare their findings with cohorts from other geographies to establish whether Kashmiri women with PCOS carry distinct gut microbiota patterns as posited from the introduction of this manuscript.
Specific comments
Line 144: typo? “stored at -80 C until analysis for further analysis”.
Lines 172-178: more detail needed for section on 16S rRNA gene sequence data processing. What pipeline was used- how were sequences quality filtered, how were paired end data managed, denoised, checked for chimeras etc.
Line 175: the greengenes database has not been updated since 2013. Authors should use actively maintained 16S rRNA sequence databases e.g. SILVA
Line 183: was a multiple comparisons adjustment applied to the mann whitney tests?
Line 198: can authors explain how and why read depth is included in the models? This is accounted for if using total sum scaled relative abundances or log ratio transformations of counts data.
Line 208, 210: Bray-Curtis dissimilarity not distance.
Line 223: possible formatting issue- table footnote merged into main text.
Figure 1a and 1b are difficult to read and authors provide minimal interpretation of these figures. For example, how do these genera and family classifications relate to Firmicutes, Bacteroidetes, Proteobacteria ranks mentioned in line 227? I would suggest to clearly indicate what C and P labels represent, and arrange bars by ascending/descending relative abundances within the cohorts (for example see figure 1 in https://microbiome.github.io/tutorials/Composition.html).
Line 246: Average relative abundance of Bifidobacterium genus is 7.7% in PCOS cohort (line 246), but Bifidobacteriaceae family is 7.63% (line 251). Please double check numbers.
Section 3.3: Why were hormonal profiles combined for all study participants and not compared between PCOS vs control cohorts?
Following from above comment, since FSH levels were not significantly different between PCOS and control (Table 1) then what is the significance/implication of the Eubacterium and FSH association highlighted here?
Figure 2h: missing values on vertical axis
Figure 2i: please double check negative Shannon diversity index values
Figure 2 legend: typo “familia”
Line 346-370: I found this paragraph on gut bifidobacteria and PCOS inadequate. Is lactose intolerance the sole driving factor of bifidobacterial abundance in the human gut? What is the subsequent link between gut bifidobacteria and PCOS, can authors speculate on possible mechanisms? How does the association with ALP tie into PCOS?
Lines 391-397: The discussion on Eubacterium does not mention PCOS, as such it is unclear how this taxon is relevant to PCOS.
Reviewer 2 Report
Introduction
- Line 72- citing Europeans and Chinese- women? Needs citation(s).
- Lines 71-78 may all related to #28 citation, but #28 needs to be cited earlier- in line 71.
- Line 78- “The gut microbiota alterations in humans associate with obesity29,30” not a full sentence. Please complete the thought.
- Line 96- here it would be nice to state why authors think PCOS may cause change in the gut microbiome- PCOS affects multiple systems? PCOS is inflammatory? Please expand upon the hypothesis.
Methods
- Line 121- Suggest removing the ‘rd’ and ‘th’ from days 3-7.
- Line 121- what samples are authors referring to here? Blood or stool? Please be specific.
- Line 127- ‘ is in wrong place- please place it between the t and s of participants.
- Line 139 and Line144- one states sera and one states plasma- which was collected??
- Stool sample collection criteria are VERY specific and demand participants to donate samples in the facility, at a certain time, specific amount…. What is patients could not donate samples at that time and place? Seems to be difficult for participants.
- 16S is not a great way to identify microbiome profiles in humans because human microbiota are highly variable. A better (but also more expensive method) is shotgun sequencing. Please address justification for 16S. or maybe discuss as a limitation.
- Were any women with PCOS on any drug treatment regimes? NSAIDS?
- Line 208- please add ‘alpha’ before diversity as Shannon’s index is an alpha diversity (intra-individual) output.
Results
- Figure 1a and 1b are pretty, but they aren’t very useful for interpreting the data . Suggest deleting them or adding them to supplemental materials.
- Line 258-264- Are the associations correlations? How were the associations calculated?
- Figure 2- This is confusing- are these plots showing both genera and familia? That’s what the caption states, but which is which? Why are H and I labeled the same? Some Y axes are labeled with relative abundance, but others are not. Maybe better labels are needed. It is not interpretable currently. Please redesign.
- Paragraph 3.2- there are no italics used for bacterial names. They are italicized in 3.4.
- The case v. control is confusing- can the difference be restated in the results?
- It is nice to have a brief sentence identifying the figure in the captions. Figure 3 needs to be revised. Also the case points are highly variable.
Discussion
- Line 313- 39 patients ended the study. Please state where the 1 participant left since 40 total were recruited.
- It is not clear why authors highlight Bifidobacterium when other genera are discussed in the paper. This comment is based on Table 2 and Figure 2. Please provide more discussion on this.
- Please be consistent with italics.
- The use of ‘association’ when discussing statistics is confusing.
Round 2
Reviewer 1 Report
2.Line 144: typo? “stored at -80 C until analysis for further analysis”.
We corrected the typo, line 144.
New comment: Sentence structure still needs slight correction- “was stored at -80°C until for further analysis”.
4.“Line 175: the greengenes database has not been updated since 2013. Authors should use actively maintained 16S rRNA sequence databases e.g. SILVA”
We would like to clarify this: The database used was not the 2013 version of Greengenes but the curated version of Greengenes by Illumina, which is available through Illumina Basespace (under the description of the 16s Metagenomics app, https://www.illumina.com/products/by-type/informatics-products/basespace-sequence-hub/apps/16s-metagenomics.html ) and allows the OTU analyses of 16s under the manufacturers (Illumina) directions. In other projects we have evaluated SILVA, which is provided here as an example by the Reviewer, and found that to achieve better results one needs to curate SILVA manually before using it. Thus, we chose the Illumina curated version of Greengenes that provides consistent and reliable results down to the genus level in our experience, while at the species level with 16s it seems to not classify roughly 30% of reads (comparable to SILVA). Thus, in this paper we focused on genus and family taxonomy levels.
New comment" I understand that while it can be data set specific, can authors provide metrics, figures or references showing that the Illumina-curated greengenes achieves better results compared with SILVA?
5.“Line 183: was a multiple comparisons adjustment applied to the mann whitney tests?”
No multiple testing comparisons adjustment was applied to the Mann-Whitney U tests because this was an exploratory analysis, which was further supported by the more advanced analysis using MaAsLin where we accounted for the multiple testing with the Benjamini-Hochberg (BH) method. However, we looked for consistency between these two analyses and for example excluded one genus, Alkalibacterium from our main findings as the effect directions were not consistent for Alkalibacterium between Mann-Whitney U-test and MaAsLin with the multiple test correction (Q<0.25). We have rephrased the text to avoid misunderstanding in lines 258-261 and the legend of Table 2.
New comment: I disagree with the reasoning that no adjustment for multiple testing was applied because this was an exploratory analysis. Authors can compare adjusted mann whitney results with those from maaslin.
6.Line 198: can authors explain how and why read depth is included in the models? This is accounted for if using total sum scaled relative abundances or log ratio transformations of counts data.
Read depth is a potential confounder when comparing abundance data as samples with shallow sequencing will have less chance of measuring taxa of lower abundance, thus introducing bias for non-null measured taxa, due to the compositionality of the relative abundance data. One way to overcome differences in read depth is rarefying, however we did not opt to rarefy as it throws out the sequence data. Instead, we included read depth as a covariate in the MaAslin analysis, as a continuous covariate when comparing total sum scaled relative abundances. This was particularly taken care of because the read depth had much more variability in the controls than in cases (see Table 1). We also had excluded one outlier sample due to low number of reads (already mentioned in methods) prior to the statistical analysis.
New comment: Since total sum scaled relative abundances accounts for read depth, there is no need for including read depth as covariate. Can authors show how the maaslin analysis changes with/without read depth as a covariate?
9.”Figure 1a and 1b are difficult to read and authors provide minimal interpretation of these figures. For example, how do these genera and family classifications relate to Firmicutes, Bacteroidetes, Proteobacteria ranks mentioned in line 227? I would suggest to clearly indicate what C and P labels represent, and arrange bars by ascending/descending relative abundances within the cohorts (for example see figure 1 in https://microbiome.github.io/tutorials/Composition.html). “
We thank the Reviewer from this observation. We believe the Krona plots were not initially correctly pointed out in the text and may have been missed by the Reviewers. We have reorganised the text and would like to point the reviewer to the Krona plots (FigureS1A and S1B) where all assigned OTUs are visible separately for cases and controls, including a clear domination of the Firmicutes enterotype. We have also removed previous Figures 1a and 1b to the supplementary, as suggested by Reviewer 2. We believe that our Krona plots together with the bar plots provide the information at the level suggested by the Reviewer (the link above).
New comment: While I would prefer to see a main text figure since differences in community composition is a main finding of this study, will interactive krona plots be available as supplementary to readers if the article is published? Authors can consider ordination plots coupled with say linear discriminant analysis to visualise compositional differences between PCOS and control samples.
10.“Line 246: Average relative abundance of Bifidobacterium genus is 7.7% in PCOS cohort (line 246), but Bifidobacteriaceae family is 7.63% (line 251). Please double check numbers.”
We double checked where the difference comes from. The family Bifidobacteriaceae has additional three genera (Scardovia and two unclassified, visible in FigureS1A and S1B). Therefore the mean relative abundance of Bifidobacterium and Bifidobacteriaceae are slightly different in every sample.
New comment: I should have pointed this out more clearly- I meant to double check the values themselves as they indicate that there is more Bifidobacterium (7.7%) than Bifidobacteriaceae (7.63%) when Bifidobacterium should be a subset of Bifidobacteriaceae.
12.“Following from above comment, since FSH levels were not significantly different between PCOS and control (Table 1) then what is the significance/implication of the Eubacterium and FSH association highlighted here?”
FSH (Follicle Stimulating Hormone), testosterone, and LH (luteinising hormone) show imbalance in PCOS. It is a common and valuable approach to use continuous outcomes in epidemiological research, as small increases (below clinical threshold) in these outcomes may point to etiological pathways when sample sizes are combined. That is why we checked for association with FHS, LH, LH/FSH and testosterone (TableS4, 5,6,7,8,9,10 and 11) at the genus and family level and used multiple testing correction to avoid false positive associations. We have expanded on the discussion regarding the relevance of this association we detected. Please see the updated discussion on lines 464-480.
New comment: I understand that these hormones are usually measured in PCOS. However, since there is no statistical difference in hormone levels in this cohort, what is the significance/implication of the association between Eubacterium and FSH? From these data, we cannot infer that Eubacterium is associated with FSH since FSH is not different between cohorts in the first place.
16. Line 346-370: I found this paragraph on gut bifidobacteria and PCOS inadequate. Is lactose intolerance the sole driving factor of bifidobacterial abundance in the human gut? What is the subsequent link between gut bifidobacteria and PCOS, can authors speculate on possible mechanisms? How does the association with ALP tie into PCOS?
Bifidobacterium is an important genus for host health, associated with favourable outcomes, especially in the early years of life. Lactose intolerance in combination with lactose intake is so far the only confirmed driving factor of Bifidobacterium abundance in adults. We had earlier showed a genetic link between family history of diabetes and Bifidobacterium abundance (Kurilshikov. et al, 2021). In our current research, for the first time we see a link through liver function, suggesting systemic effects of the genus on host homeostasis. We added this information in the discussion.
New comment: Thank you for addressing my comment, however, I do not see how the link between Bifidobacterium and liver function is explicitly associated with PCOS. Can authors provide a specific inference of how this may be, with literature to support any inferred mechanisms? As the finding of increased Bifidobacterium in PCOS women forms the title of the manuscript, I feel that authors should discuss this aspect in greater detail and provide a way forward with their results. How do these findings inform future directions?
17.Lines 391-397: The discussion on Eubacterium does not mention PCOS, as such it is unclear how this taxon is relevant to PCOS.
We added an explanation on this association, we do not see a direct link between Eubacterium and PCOS.
New comment: If Eubacterium is not linked with PCOS, why highlight this species?
Author Response
Please see the attachment.

This manuscript is a resubmission of an earlier submission. The following is a list of the peer review reports and author responses from that submission.
Round 1
Reviewer 1 Report
The authors present a study on gut microbiome diversity in Indian women with PCOS. With this paper they contribute to the data emerging from this complex field of research.
I have a few major comments:
The authors write in the introduction that gut microbiome alterations are associated with obesity, insulin resistance and glucose metabolism. Yet, the authors chose to include a PCOS population with a normal BMI but impaired fasting glucose levels. It is not clear to me why the authors chose to include such as study population. In my opinion, this study should be performed in a normal weight PCOS population without any metabolic derangements to study the possible ‘effect’ of PCOS on the composition of gut microbiome – and not the possible effect of weight/obesity, insulin resistance or impaired glucose tolerance.
Fasting glucose levels were also increased to a prediabetic level in the control population. Do the authors think this population is fit as ‘healthy controls’?
Finally, the authors should justify the use of transabdominal ultrasound in the diagnosis of PCOS, taking into account the findings of the following paper ‘Definition and significance of polycystic ovarian morphology: a task force report from the Androgen Excess and Polycystic Ovary Syndrome Society’ by prof Dewailly et al. in Human Reproduction Update 2014.
The rationale why the authors chose to analyse associations between quantitative traits and prolactin levels and between FSH levels is not clear to me. Perhaps, the authors could comment on these findings in the discussion?
The authors may consider to show table 2 and 3 in supplement. In my opinion, the figures are more informative for this type of results.
Reviewer 2 Report
The authors examine profiled of gut microbiome in faecal samples of Kashmiri women with polycystic ovary syndrome. The topic of the study is very intresting, however the sample of participants is very small. The main limitation of the study is BMI- the PCOS group are overwaight whereas control are lean therefore it is hard to compare this to gropus.
The Tample is not correct- should be used for 2021. There is no ethical statement
Manuscript should be improved:
Abstract there is no information about metabolic parameters which was examine, there is no p values. Abstract is not clear, conclusion should be based on findings.
Introduction:
- it is not true that polycystic ovary is diagnosed based on transabdominal ultrasonography.
- There are mistakes in the References e.g. in numer 4 is not any information about prevelance of PCOS it is article about autoantibodies in PCOS.
- There is statement about pathogenesis of PCOS, however this should focus rather on obesity, insulin resistance and gut microbiome (lines 49-50).
Material and methods:
- It is not clear why authors choose only women with age 16-25, mayby should examine older than 18 ? It is not clear why women under 16 was hospitlized in the clinic for adults?
- How was diagnose hirsutism? What does it mean male pattern hair growth, androgenic alopecia?
- Why was measured thyroxine (T4), triiodothyronine (T3) not freeT3 and free T4 of this hormones? Is there any correlation wit gut microbiome?
- It is not clear why only FSH, LH and testosterone was collected between 3 and 7 days of cycle, patient and control groups was hospitized twice to take blood for above hormones and in any day of cycle to take blood for others parameters? It is not clear why control group was hospitilized, therefore it was not control..
- There is no CVs for measurments.
- Line 164 theres is missed information
- Why day of menstrual cycle is important for parameters of stool?
- The PCOS group are overwaight whereas control are lean therefore it is hard to compare this to gropus.
Results:
It is not clear what kind of distribution have data norma lor not-normal? Information about it is in the limitation of the study.
Table 1 it is not clear what authors show mean nor median?, in the weigt, waist, hip circumference FBG etc. there is mistake in p value (should be p<0.01); there is information about blood preasure, weight, however there is no information about it in Materials and Methods section. The word: Hormonal and Biochemical Parameters should be removed and title of table should be change.
Table 1 should be diveded for 2, there is a huge differences in waist between groups, is it correct? It is not clear why control group have elevated fasting blood glucose- there are young and not-obese?
Tables: there is mistake in p value. In Tables p value shoul be presented as p=0.01 or p<0.01
It is not clear what is it Table S1?
Line 222: should be removed weight and high
Disscucion
- The authors should explain therir fingdings connected with significant direct association between butyrate producing Eubacterium and follicle-stimulating hormone levels.
- Line 290 weight should be removed
- In the citation there are mistakes e.g. references numer 45 there is no focus on PCOS but T2D, etc. Every refernces should be cheked carefully once again.
- The conclusion should be change according to the main findings.